# Multicentric study on the reproducibility and robustness of PET-based radiomics features with a realistic activity painting phantom

Piroska Kallos-Balogh[1,2]*, Norman Felix Vas[1], Zoltan Toth[3], Szabolcs Szakall[4], Peter Szabo[5], Ildiko Garai[1,5], Zita Kepes[1], Attila Forgacs[6], Lilla Szatmáriné Egeresi[7], Dahlbom Magnus[8], Laszlo Balkay[1,2]

1 Division of Nuclear Medicine and Translational Imaging, Department of Medical Imaging, Faculty of Medicine, University of Debrecen, Debrecen, Hungary, 2 Doctoral School of Molecular Medicine, Faculty of Medicine, University of Debrecen, Debrecen, Hungary, 3 Medicopus Healthcare Provider and Public Nonprofit Ltd., Somogy County Moritz Kaposi Teaching Hospital, Kaposvár, Hungary, 4 Pozitron-Diagnostics Ltd., Budapest, Hungary, 5 Scanomed Ltd., Debrecen, Debrecen, Hungary, 6 Mediso Medical Imaging Systems, Budapest, Hungary, 7 Division of Radiology and Imaging Science, Department of Medical Imaging, Faculty of Medicine, University of Debrecen, Debrecen, Hungary, 8 Ahmanson Translational Theranostics Division, Department of Molecular and Medical Pharmacology, David Geffen School of Medicine, UCLA, Los Angeles, California, United States of America

* balogh.piroska@med.unideb.hu

**Data Availability Statement:** All relevant data are within the manuscript and its Supporting information files.

## Abstract

Previously, we developed an "activity painting" tool for PET image simulation; however, it could simulate heterogeneous patterns only in the air. We aimed to improve this phantom technique to simulate arbitrary lesions in a radioactive background to perform relevant multi-center radiomic analysis. We conducted measurements moving a $^{22}$Na point source in a 20-liter background volume filled with 5 kBq/mL activity with an adequately controlled robotic system to prevent the surge of the water. Three different lesion patterns were "activity-painted" in five PET/CT cameras, resulting in 8 different reconstructions. We calculated 46 radiomic indeces (RI) for each lesion and imaging setting, applying absolute and relative discretization. Reproducibility and reliability were determined by the inter-setting coefficient of variation (CV) and the intraclass correlation coefficient (ICC). Hypothesis tests were used to compare RI between lesions. By simulating precisely the same lesions, we confirmed that the reconstructed voxel size and the spatial resolution of different PET cameras were critical for higher order RI. Considering conventional RIs, the SUV$_{peak}$ and SUV$_{mean}$ proved the most reliable (CV<10%). CVs above 25% are more common for higher order RIs, but we also found that low CVs do not necessarily imply robust parameters but often rather insensitive RIs. Based on the hypothesis test, most RIs could clearly distinguish between the various lesions using absolute resampling. ICC analysis also revealed that most RIs were more reproducible with absolute discretization. The activity painting method in a real radioactive environment proved suitable for precisely detecting the radiomic differences derived from the different camera settings and texture characteristics. We also found that inter-setting CV is not an appropriate metric for analyzing RI parameters' reliability and robustness. Although multicentric cohorts are increasingly common in radiomics analysis, realistic texture

**Funding:** The author(s) received no specific funding for this work.

**Competing interests:** The authors have declared that no competing interests exist.

**Abbreviations:** CV, Coefficient of Variation; FOV, Field of View; GLCM, grey level co-occurrence matrix; GLRLM, grey-level zone length matrix; GLZLM, grey-level run length matrix; ICC, Intraclass Correlation Coefficient; NGLDM, neighborhood grey-level difference matrix; RI, Radiomics Indices; TLG, Total Lesion Glycolysis; VOI, Volume of Interest.

phantoms can provide indispensable information on the sensitivity of an RI and how an individual RI parameter measures the texture.

## Background

Since combined positron emission tomography (PET) and computed tomography (CT) serves as a valuable means to measure the functional state of the human body *in vivo*, in a quantitative way, this hybrid modality also ensures the accomplishment of comparative patient studies. More recently, intensive focus has been placed upon investigating quantitative pattern parameters to extract latent information from PET images. This endeavor—denoted as radiomics–is gaining increasing attention in medical imaging fields, including nuclear medicine [1–5].

Despite several promising results, the field of radiomics is still attempting to reach a breakthrough in usefulness, as several previous publications highlighted fundamental doubts, mainly regarding methodology [2, 3, 6–12]. Additionally, scientific data indicate that the reproducibility and robustness of the radiomic indices, also remain unsatisfactory [13–17]. The diagnostically relevant radiomic features found in individual clinical trials most commonly do not coincide, even for the same disease [17]. To assess and improve research quality in the field of radiomics and machine learning, the "checklist for the evaluation of radiomics research" (CLEAR) guideline, and the "methodological radiomics score" (METRICS) quality scoring tools have recently been published with the support of the European Society of Medical Imaging Informatics (EuSoMII) organization [18, 19]. CLEAR aims to set a standard to improve the quality, and subsequently the reproducibility of radiomics research presentation [18]. Including nine major categories, however, the Metrics is a scoring tool used to evaluate the methodological quality of radiomics studies [19]. The first four subclasses were in descending order of importance the following: "study design", "imaging data", "image processing", and "feature extraction".

In case of a PET camera, besides the technical design of the device, the image quality is mostly influenced by the applied study protocol and the (pre)-defined parameters. Including amongst others the injected activity, acquisition time, the type and the parameters of the reconstruction algorithm, the image matrix size, and the type of the post-filters. Consequently, the need to explore the effects of different PET devices on the accuracy of the radiomics calculations and to support concept harmonization, especially in a multi-device study, is warranted [20–22]. Each scanner model possesses specific performance and diagnostic acquisition protocols, which adumbrate the variable distortions of the reconstructed data. A possible way to overcome this limitation is to perform phantom data-based harmonization prior to the accomplishment of human studies [23–26]. Although, this approach assumes a phantom with a highly reproducible uptake pattern.

In recent years, great advancements have been made in the design of novel phantom for this purpose. For example, a noise-based image harmonization has been introduced [27] utilizing acrylic rods with different fill densities. Bead type phantom has also been developed [28] to analyze radiomics indices and their power to discriminate different heterogeneity. Moreover, 3D printed fillable inserts have been constructed based on Non-Small-Cell-Lung-Cancer tumors extracted from patient studies [29]. In this paper the authors demonstrated that the reliability of the radiomic features depends on the nature of the underlying data, including tumor size, shape, tracer uptake level, contrast, and intratumor radiotracer accretion. It was also established that the image reconstruction methods and settings, the noise, the

discretization, and the delineation methods largely influenced the repeatability of 2-[18F]-FDG PET radiomic features. Additionally, other heterogeneous phantoms were designed using acrylic beads to simulate inhomogeneous patterns in the field of view (FOV) of the PET device [30]. Nevertheless, the major drawback of these phantoms is that they are not applicable to create textures of any arbitrary shape and heterogeneity, moreover technical know-how is required to prepare such phantom types.

To bypass these limitations, our research team has developed a new phantom technique suitable for evaluating and comparing the performance of different scanners in radiomics [31]. Herein, we aimed to analyse and demonstrate the effect of differences in PET camera configurations on the visual appearance of the lesions and radiomic features using realistic heterogeneous phantom data. In addition, by implementing the activity painting method in background radioactivity to simulate different lesions, we have created a more realistic and general-purpose PET radiomics phantom. In this work, the term simulation will always be used to refer to the process of activity painting.

## Materials and methods

### Phantom design

The original activity painter phantom design was described in the previous work of our research group [31]. Briefly, it was built applying three linear stages (Zaber Technologies, T-LSM050A motorized stage, Fig 1B) that provided the controlled movement of a 22Na spheric point source (Eckert Ziegler Isotope Products, Inc.; 0.25 mm diameter, with activity of 1.1 MBq,) in the FOV of the PET camera. Utilizing Matlab code-based controlling, the movement was converted into a 3D activity distribution on the reconstructed data.

To ensure experimental conditions similar to real-time PET imaging, in the present work the movement of the point source takes place in a 20-liter water tank containing background radioactivity (Fig 1).

The water tank was filled with approximately 15 liters of 2-[18F]-FDG solution (15.06 L ±0.894) to represent scattering, attenuation media, and activity outside the FOV. Using the

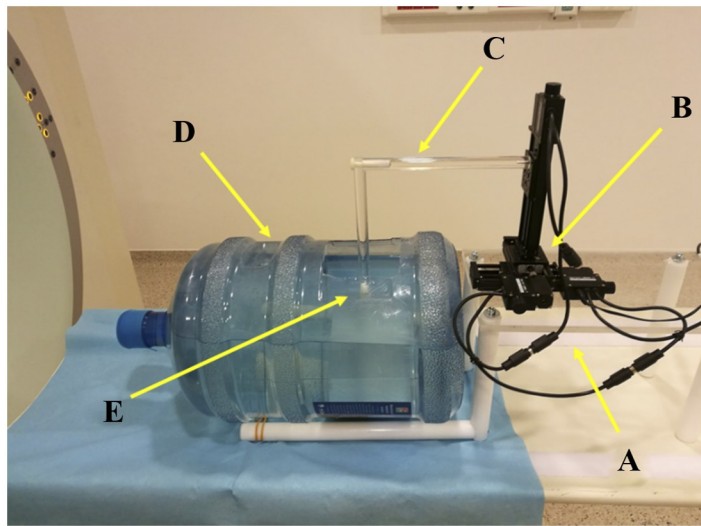

**Fig 1. Positioning system on a scanner bed.** The system consists of a plastic holder (A), linear stages (B), an L-shaped plastic rod (C), a 20-liter water tank (D), and a 22Na point source (E).

**Table 1. Detailed characteristics of the measurement.**

|  |  | Siemens Biograph True Point 64 | Mediso AnyScan PET/CT | GE Discovery IQ | Philips Gemini TF 64 | GE Discovery MI |
|---|---|---|---|---|---|---|
|  | Volume [L] | 15.1 | 15.0 | 15.2 | 15.0 | 15.0 |
| Lesion 1 | Initial Activity [MBq] | 87.2 | 88.5 | 88.1 | 89.4 | 89.7 |
|  | Acquisition Time [sec] | 1730 | 1730 | 1730 | 1730 | 1730 |
|  | Time inaccuracy [sec] | -20 | -20 | -4 | -8 | -9 |
| Lesion 2 | Initial Activity [MBq] | 70.8 | 72.6 | 71.9 | 72.5 | 73.3 |
|  | Acquisition Time [sec] | 1883 | 1883 | 1883 | 1883 | 1883 |
|  | Time inaccuracy [sec] | -1 | -10 | -4 | -10 | -7 |
| Lesion 3 | Initial Activity [MBq] | 55.8 | 57.6 | 57.1 | 58.1 | 59.1 |
|  | Acquisition Time [sec] | 2600 | 2600 | 2600 | 2600 | 2600 |
|  | Time inaccuracy [sec] | -18 | -8 | -4 | -6 | -2 |

Data on the volume of the water and activity at the beginning of the simulation, acquisition time and time inaccuracy in case of all scans.

plastic holder and the L-shaped rod, we were able to elevate the linear stages and submerge the $^{22}$Na point source in the water, thereafter, we performed the positioning inside the water tank. A cylindrical rod with a uniform cross-section was applied to minimize wave propagation during the movement. All measurements were performed in 1 FOV, the activity in the water was set to scan start, and we intended to maintain the activity at a similar value (88.50 MBq±0.99 SD). The volume of the water and the activity concentration were set before every scan (as seen in Table 1).

## Lesions

The activity painting phantom simulated three different human lesions downloaded on 17 March 2018 from an online open-access database [32]. Given the anonymized nature of the database, the authors had no means of uncovering patient data. Even though this database contains seven lesions, only those were selected that demonstrated different degrees of visual heterogeneity (Patient 1, Patient 3, and Patient 5, seen in Fig 2). The volumes of Lesion 1 to 3 were 8.66 mL, 11.55 mL, and 15.44 mL; respectively. Since the current linear stages are only able to cover the size of a 5x5x5 cm$^3$ cube, the lesions had to be resized to fit this dimension.

## Scanners and reconstructions

Data were acquired with the application of five different PET/CT scanners (up-to-date cameras and older ones) utilizing different reconstruction settings (as shown in Table 2). Since we used two different reconstruction settings in the case of three PET/CT devices, a total of eight distinct imaging settings were defined. In the following, we refer to these settings with letters A-H. The selected PET/CT systems and their characteristics are detailed in the previously published works of Bettinardi et al., Hsu et al., Jakoby et al., and Surti et al. [33–36].

 The water volume was determined by measuring the weight of the water-filled tank using a scale, and the activity concentration was calculated before each scan. In order to adjust the time of PET data collection to the time of the simulations, it was essential to know the time of the simulations for all three lesions. The required acquisition time was estimated by the performance of the activity painting process five times for each lesion, placing the apparatus (the linear stages) in a table without radioactivity. Then, the estimated simulation times were calculated as the average of the five consecutive activity painting process times. Furthermore, the actual time for each lesion simulation was also recorded, and the timing inaccuracy of each

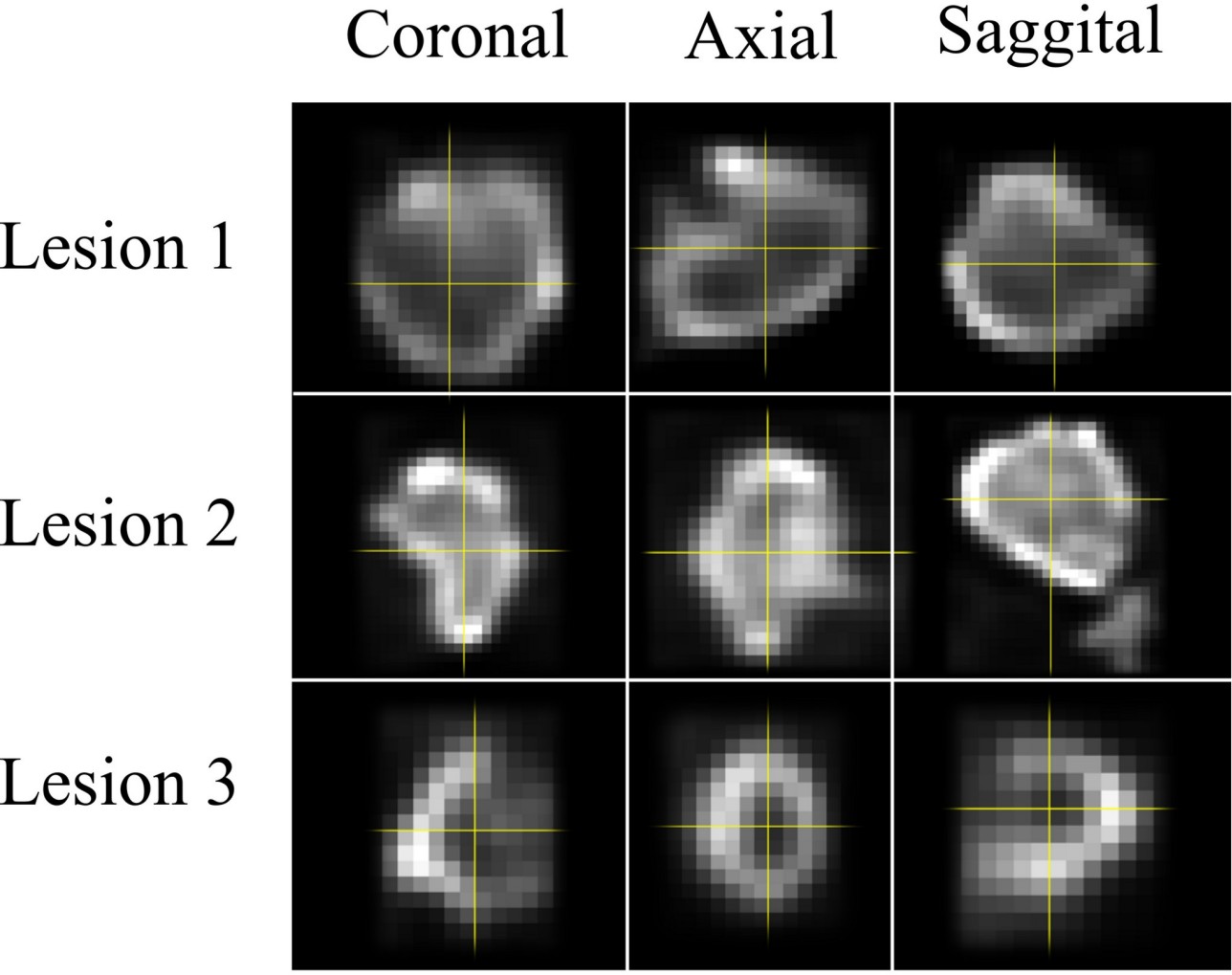

**Fig 2. Lesions selected from the online database.** Lesion 1, Lesion 2, and Lesion 3 in different orthogonal orientations. Since the current linear stages are only able to cover the size of a 5x5x5 cm$^3$ cube, the lesion sizes had to be customized according to this dimension.

activity painting process was calculated as the percentage difference between the actual and estimated simulation times.

### Segmentation

All image processing steps, including segmentation, discretization, and the extraction of the radiomic features were performed using the LIFEx v4.34 software [37]. Although this tool allows manual segmentation as well as several threshold-based segmentations (n, n%, Nestle, peak), we presently opted for the former option.

### Discretization

Followed by segmentation, the following two different discretization methods were applied: relative and absolute discretization. Although these methods have several characteristics in common, some differences must be addressed. The relative method means the clustering of

**Table 2. PET/CT scanners and the reconstruction settings.**

| Code | Scanner Model | Reconstruction |
|------|---------------|----------------|
| A | Siemens Biograph TruePoint 64 | 2D OSEM[a] 3i8s, (2.67 mm isotropic), Gaussian Post (FWHM[b] 5mm) |
| B | | 2D OSEM[a] 3i8s, 5mm (4x4x3 mm$^3$) Gaussian Post (FWHM[b] 5mm) |
| C | Mediso AnyScan PET/CT | TT3D[c] 6i6s (3 mm isotropic) No post-filtering |
| D | GE Discovery IQ | VUE Point HD[d], 4i12s (2.73 mm isotropic) Z-Axis filter: standard, SharpIR[e], Gaussian Post (FWHM[b] 6.4 mm) |
| E | | VUE Point HD[d], 4i12s (3.64 mm isotropic) Z-Axis filter: standard Gaussian Post (FWHM[b] 4.4 mm) |
| F | Philips Gemini TF 64 | 3D RAMLA[f] (4 mm isotropic) No post-filtering |
| G | GE Discovery MI | Q. Clear[g], beta 450 (2.73 mm isotropic) SharpIR[e], Z-Axis filter: none |
| H | | 3i8s VPFXS[h], SharpIR[e] (2.73 mm isotropic) Z-Axis Filter Light, Filter cutoff(mm) 3.2 |

The PET/CT modalities, the applied reconstruction settings, and the filters are presented. Two different reconstruction modes were chosen for three scanners (Siemens Biograph TruePoint 64, GE Discovery IQ and GE Discovery MI). Image voxel sizes are shown within parentheses. The "n" iteration number and "m" subset number are labelled "nims".

[a]2D OSEM: Two-Dimensional Ordered Subsets Expectation Maximalization.

[b]FWHM: Full Width at Half Maximum.

[c]TT3D: Three-Dimensional Time-of-Flight Tomographic Reconstruction.

[d]VUE Point HD: a branded term created by GE Healthcare to denote their advanced high-definition reconstruction technology.

[e]SharpIR: GE Healthcare's proprietary PET image reconstruction algorithm with resolution recovery and point spread function modeling.

[f]3D RAMLA: Three-Dimensional Reconstruction Algorithm for Maximum Likelihood Analysis,

[g]QClear: GE Healthcare's proprietary PET image reconstruction algorithms using Bayesian penalized likelihood reconstruction.

[h]VPFXS: GE Healthcare's PET reconstruction algorithm.

pixels into a fix number of bins, while the second one, the absolute discretization method uses a fixed bin size. Recent guidelines [38] refer to the relative and absolute methods with the abbreviations FBN and FBS respectively. The following formulas define the FBS and FBN methods:

$$I_{FBS}(i) = \left[ \frac{I(i)}{B} \right]$$

$$I_{FBN}(i) = \begin{cases} \left[ D\dfrac{I(i) - I_{min}}{I_{max} - I_{min}} \right] + 1 & I(i) < I_{max} \\ D & I(i) = I_{max} \end{cases}$$

where I(i) is the original value of the voxel i, and $I_{max}$ and $I_{min}$ are the maximum and minimum values of a given lesion; respectively. D refers to the number of the bin parameters, while

B indicates the bin width. Corresponding to the literature data, the B and the D values were set to 0.3125 and 64; respectively [39].

## Radiomic parameters

First order statistics (Metabolic Active Tumor Volume (MATV), Maximum Standardised Uptake Value ($SUV_{max}$), Minimum Standardised Uptake Value ($SUV_{min}$), Mean Standardised Uptake Value ($SUV_{mean}$), Standard Deviation of Standardised Uptake Value ($SUV_{std}$), Peak Standardised Uptake Value ($SUV_{peak}$), Total Lesion Glycolysis (TLG = $SUV_{mean}$ ·Volume), Segmented Shape (Volume, Sphericity and Compacity) and Histogram parameters (Skewness, Kurtosis, Entropy and Energy), and higher order parameters (based on Gray Level Co-occurrence Matrix, Grey Level Run Length Matrix, Neighborhood Grey-Level Difference Matrix and Grey-Level Zone Length Matrix) were calculated. On the whole, 46 parameters (see S1 Table) were analyzed and computed using the LifeX software, out of which $SUV_{max}$, $SUV_{min}$, $SUV_{mean}$, $SUV_{std}$, $SUV_{peak}$, TLG and Volume were considered conventional PET parameters. Overall, eight different image sets were obtained from each lesion based on the reconstruction settings (shown in Table 1).

## Statistical analysis

Differences regarding a given radiomics parameter were characterized by the comparison of the obtained values of the different imaging settings (Table 1) with the RI value of setting D (the choice of reconstruction method D was completely arbitrary, it was just a matter of choosing one of the imaging settings as a reference). The deviations were measured by relative difference (RD) expressed in % with the following formula:

$$RD = \frac{|X_i - X_D|}{X_D} * 100.$$

X indicates the given RI parameter, $i = \{A, B, C, E, F, G, H\}$, and $X_D$ is the RI value in case of imaging setting D (the reference setting).

To compare the RIs among the three lesions, non-parametric Wilcoxon rank-sum tests were also performed with the 'ranksum' Matlab function. Due to the multiple comparisons the Benjamini-Hochberg false discovery rate (FDR) method was used to correct the p-values calculated by the Wilcoxon test. The FDR-based control is less stringent than the Bonferroni correction method, however it greatly improves the power of statistical inference.

## Coefficient of variation

Inter-setting coefficient of variation (CV) was determined to examine the variability of each radiomics parameter ($RI_i$, i = {1, 2, . . ., 46} indicating the 46 different RI) on the imaging settings using the following equation:

$$CV_i = \frac{STD(RI_i)}{mean(RI_i)} \cdot 100,$$

where *STD* and *mean* are the standard deviation and the mean of an $RI_i$ from the eight imaging settings; respectively. Since we calculated CVs based on three lesions and two discretizations, a total of 276 CVs were determined. According to the CV, the radiomic parameters were characterized either by low variability (CV less than 10%) or moderate variability (CV between 10% and 25%).

### Intraclass correlation coefficient

To test the reproducibility of the radiomics parameters, Intraclass Correlation Coefficient (ICC) value was calculated for each of them based on absolute agreement, two-way mixed effects model [40].

$$ICC = \frac{MS_R - MS_E}{MS_R + (k-1)MS_E + \frac{k}{n}(MS_C - MS_E)},$$

where $MS_R$, $MS_E$, and $MS_C$ indicate the mean square for rows, the mean square of error, and the mean square of columns; respectively, $n$ refers to the number of the lesions, and $k$ is the number of the measurements.

ICC was determined by averaging the eight results obtained from the different imaging settings in the case of all lesions. Based on the acquired ICC values, the radiomics parameters can be sorted into the following categories: features with excellent (ICC > 0.9), good (0.75 < ICC ≤ 0.9), moderate (0.5 < ICC ≤ 0.75), and poor (ICC ≤ 0.5) repeatability.

## Results

### Comparison of the reconstructed images

Fig 3 demonstrates the reconstructed images of Lesions 1–3. Each column shows the three orthogonal views of the same reconstructed volume, and each row is related to the different imaging settings marked with the capital letters according to Table 2.

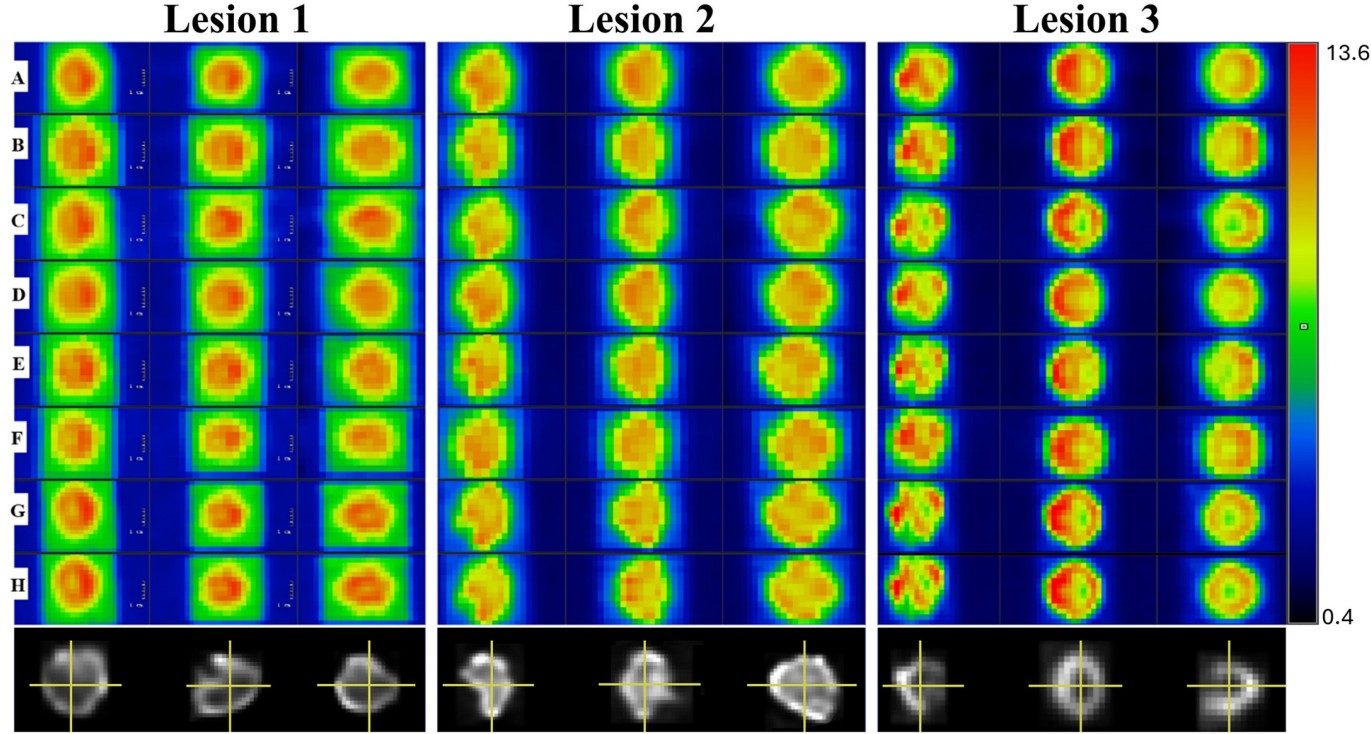

**Fig 3. Reconstructed views of the lesions in three orthogonal views.** The format and the labelling are described in the main body of the manuscript. The images were obtained with the same window level and width to ensure uniformity in visualization. The values of the colorbar are expressed in SUV. In the last row the corresponding original lesions are depicted.

The lesions were well identified since the chosen phantoms and the reconstructed images had a similar pattern. Based on the shapes of the original lesions, imaging options G and H (which belonged to the digital PET) provided the most intricately detailed imaging with the highest level of accuracy. In addition, the same reconstruction methods (G and H) supplied most of the details in case of all three lesions. Interestingly, an additional scanner with two different reconstructions (D and E) presented significantly different contrast resolutions. Moreover, the voxel size and the resolution of the PET system influenced the visual appearance and contrast of the images as well.

## Comparison of the radiomics parameters

The relative difference values between the radiomics parameters of the distinct imaging settings (A, B, C, E, F, G and H relative to D) are shown in Fig 4. We arbitrarily selected imaging setting D as the reference in the relative difference formula.

The conventional parameters show the smallest differences between the radiomics parameters of the different imaging settings; however, in some cases (TLG, Volume and $SUV_{std}$ for Lesion 1 and Lesion 2), these parameters were registered with values above 25%. As expected, the same values were recorded for most of the conventional parameters using both the absolute and the relative discretization methods for most cases. In general, we experienced that the use of absolute discretization induced greater deviations between the assessed radiomics parameters. In particular, discretization had a remarkable effect on the values of the higher-order parameters. It could be highlighted that while shape-based parameters were not considered sensitive to discretization, GLRLM and GLZLM parameters seemed to be the most reliant on the type of discretization. Comparing the results of Lesion 3 with those of Lesions 1 and 2, we observed that regarding Lesion 3, the extracted radiomic parameters were less different from each other.

## Coefficient of variation

The CVs of the radiomics parameters were calculated for all lesions with both discretization methods (Fig 5).

The first order parameters were the most robust for all lesions and discretization methods. We noted that discretization did not influence the CV value of the conventional parameters. Further, $SUV_{peak}$ was found to have the same variability of 5–10% for all lesions and segmentation methods, whereas $SUV_{mean}$ and $SUV_{max}$ showed variability between 5% and 15%. As expected, the $log_2$,–and $log_{10}$-based Entropy parameters—independently of being Histogram,– or GLCM-based—had the same values. Characterizing the shape of the segmented volume of the lesion, SHAPE_Compacity and SHAPE_Sphericity also appeared to have moderate variability, as they had a CV under 20%, with the maximum value of 12%. In addition, SHAPE_Sphericity with CV value below 5%–was detected to have the lowest variability out of the shape parameters. Furthermore, the different discretization methods did not influence the volume of the segmented VOI—the SHAPE_Volume (mL). We suppose that the high variability of the TLG could be due to the high CV of the lesion volume.

Moreover, we observed periodicity in the values of some parameters. Applying relative resampling, HISTO_Energy could be characterized by a value smaller than 5%, which was greater than in case of the application of absolute resampling. This parameter also presented periodicity. The CV values of both Histogram,—and Entropy-based parameters obtained with absolute resampling were smaller (0%-15%) compared to those measured using relative resampling (>25%). Additionally, the GLRLM_LRE showed the same periodic behavior as described earlier. Finally, we noted that GLZLM-based parameters had the highest CV values.

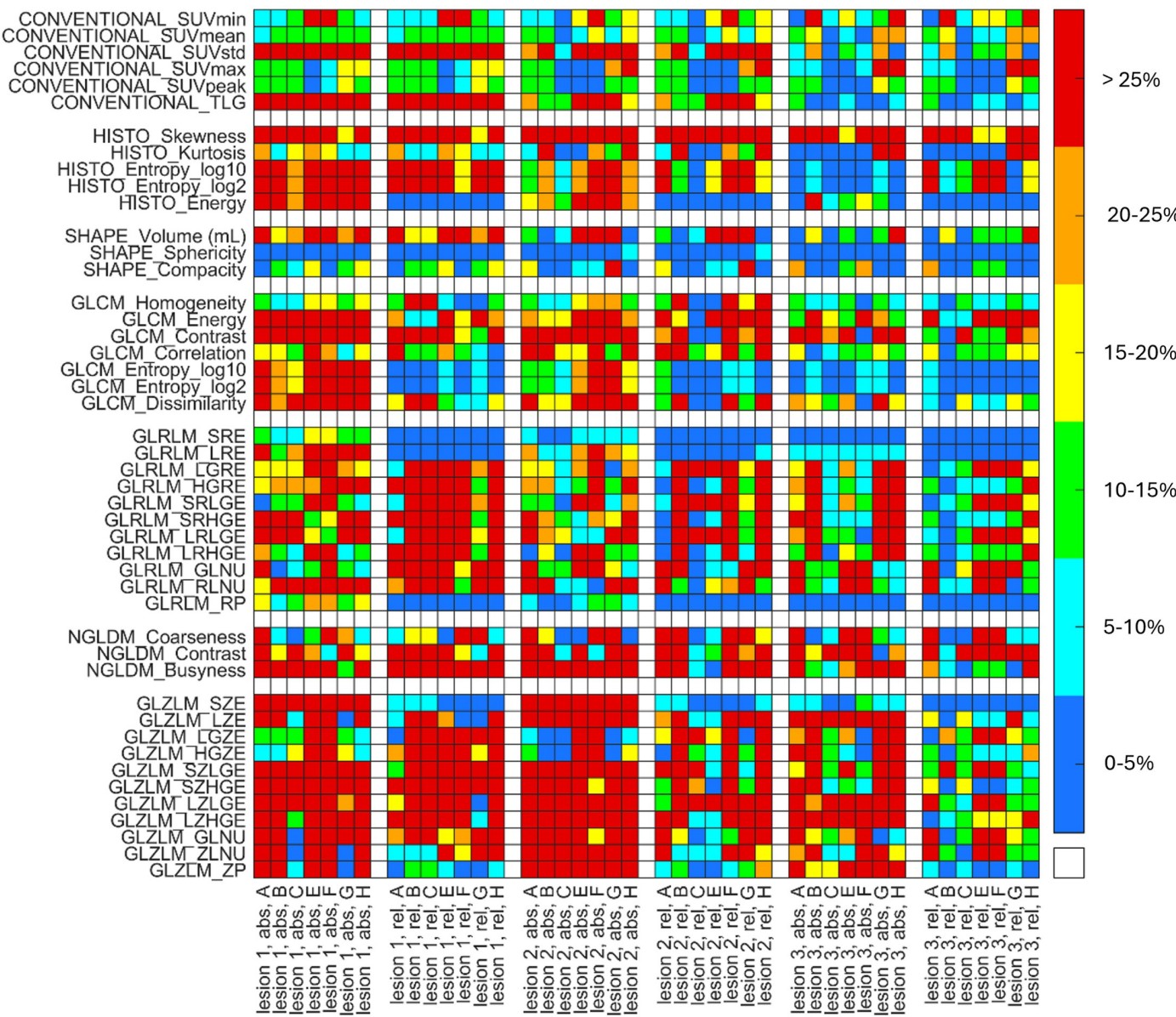

**Fig 4. Relative difference values between the radiomics parameters of the applied imaging settings.** The horizontal axis represents the imaging settings (without reference D) and the discretization methods for each of the three lesions, while the 46 radiomics indices are shown on the vertical axis. The colors corresponding to the percentage level of the relative difference value are the following, blue: 0–5%, light blue: 5–10%, green: 10–15%, yellow: 15–20%, orange: 20–25% and red: >25%.

## Conventional parameters

In our study, the maximum values of the SUV data are shown in Fig 6 with respective values being 10.33% 13.62% and 7.51% for $SUV_{mean}$, $SUV_{max}$, and $SUV_{peak}$. $SUV_{peak}$ showed the best performance out of these figures, with a CV value below 8%. Values below 33.7% were detected for the conventional parameters in every case, and the highest was registered for the Volume of Lesion 1 and 2 (as seen in Fig 6). Additionally, all the other radiomics parameters had a CV below the 25% threshold. The variability of the TLG originates from the variance of the volume. Fig 6 further strengthens that the different discretization methods do not influence the conventional parameters.

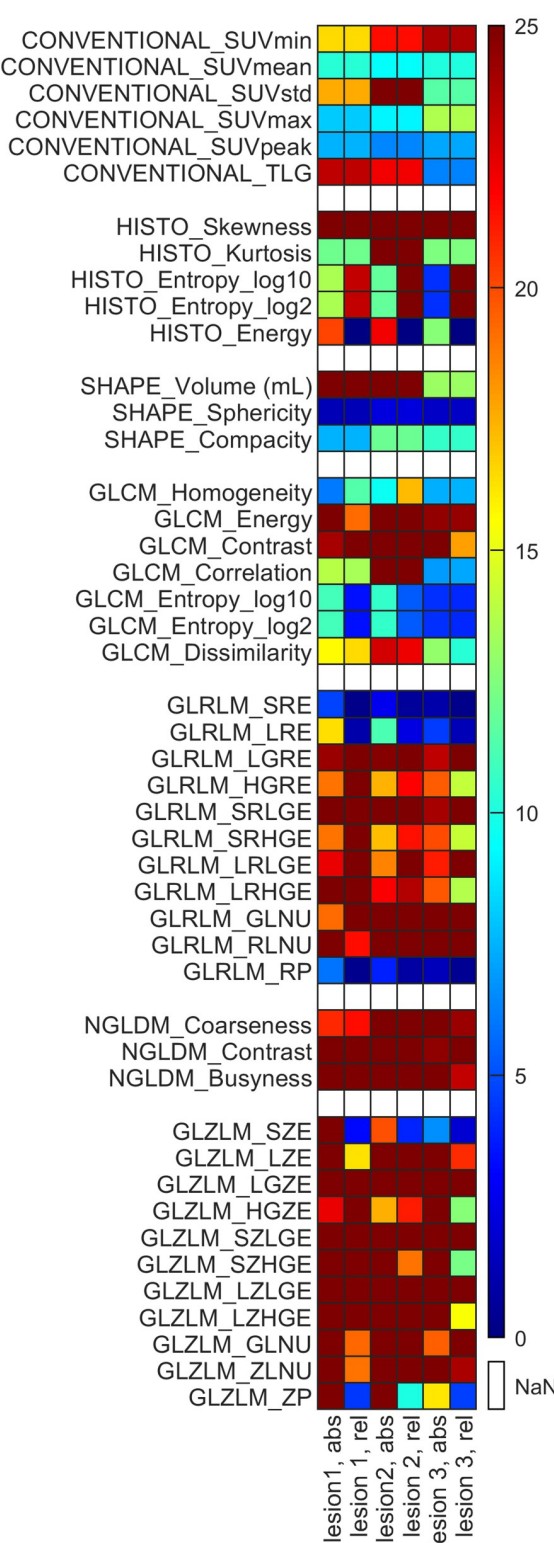

**Fig 5. Coefficient of variation values (CVs) of the Radiomics parameters.** We present CV values between 0–25%, where values <10%, 10–25%, and >25% represent low, moderate, and high variability, respectively.

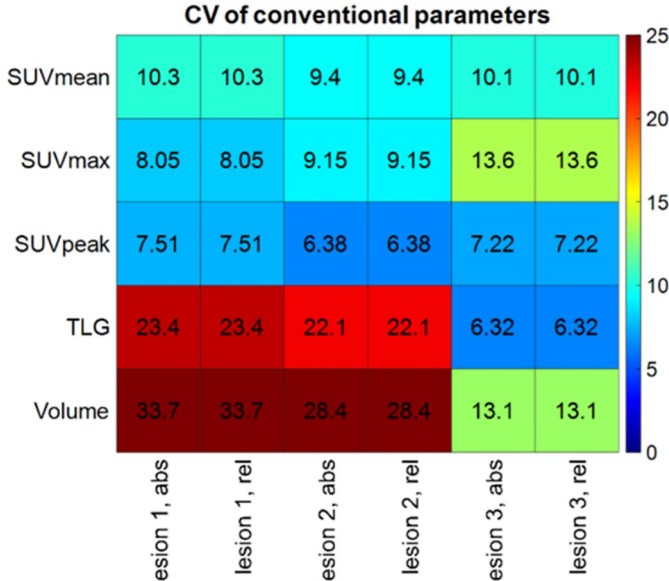

**Fig 6. Coefficient of variation (CV) values of the Conventional parameters.** The actual value of a CV is also shown in the cells.

## Spatial resampling

Spatial resampling was also performed on the images using the LIFEx software, applying 1 mm, 2.5 mm, 3 mm, and 4 mm voxel sizes (as shown in Fig 7).

Based on the findings, we can summarize that the conventional and histogram—based parameters were the least dependent on spatial resampling. $SUV_{min}$, $SUV_{std}$, TLG, HISTO_Skewness, Histo_Kurtosis, and Histo_Energy showed poor response to spatial resampling.

Among the three SHAPE parameters, Sphericity proved to be the most stable against spatial resampling effects. In addition, the CV value of Compacity improved with smaller voxel sizes. Applying spatial resampling, the CV values of GLCM_Homogeneity and GLCM_Correlation decreased, whereas the CV figures of the remaining GLCM-based parameters either stayed unchanged or increased. All GLRLM-based parameters seemed to be sensitive to spatial resampling. We recorded higher CV values for GLRLM_SRE, GLRLM_LRE, GLRLM_LGRE, GLRLM_SRLGE, GLRLM_SRHGE, GLRLM_LRLGE, and GLRLM_RP when the voxel size decreased. While spatial resampling reduced the CV values of NGLDM-based Coarseness, no noticeable change was detected in other cases. GLZLM-based parameters were also considerably affected by spatial resampling; however, with the reduction of the voxel size, no definitive tendency was observed concerning the change of their CV values.

## Wilcoxon signed-rank test

Followed by data collection from all cameras, we finally ran Wilcoxon signed-rank hypothesis tests to compare all RIs of the different lesions. The calculated p-values were corrected using the Benjamini-Hochberg false discovery rate method due to the multiple comparisons. The results are displayed as a colormap in Fig 8, where the cold colors represent a statistically significant difference between the given radiomics parameters *(p<0.05)*. In contrast, the warm colors indicate no considerable disparity between the assessed figures *(p>0.05)*.

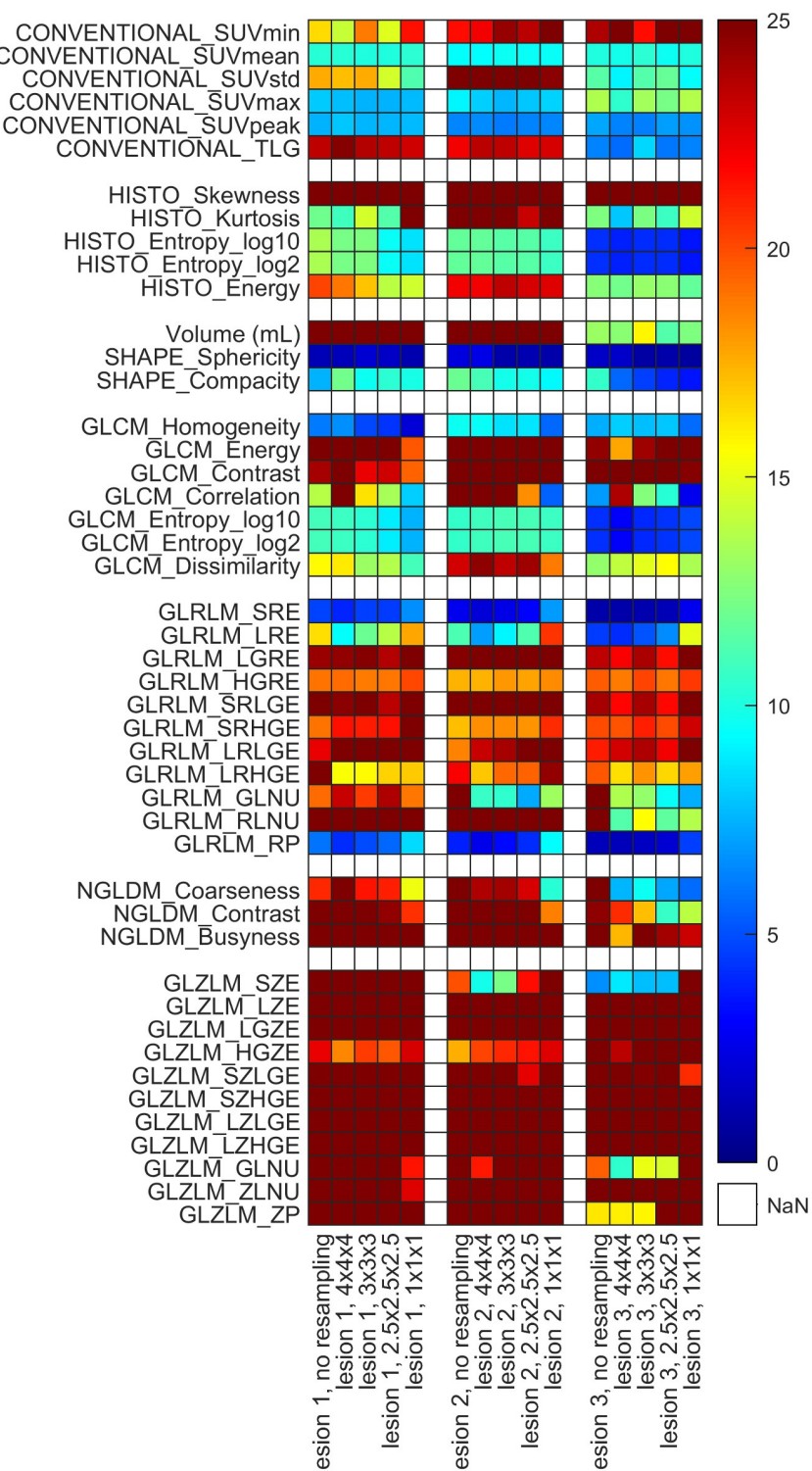

**Fig 7. Effect of spatial resampling on the radiomics parameters.** The heat map indicates the coefficient of variation (CV) of the radiomics parameters. After displaying the colors of the original data (without spatial resampling), the CVs obtained with voxel sizes of 4×4×4, 3×3×3, 2.5×2.5×2.5 and 1×1×1 mm$^3$ are shown. Here, we presented those results that were calculated using absolute discretization.

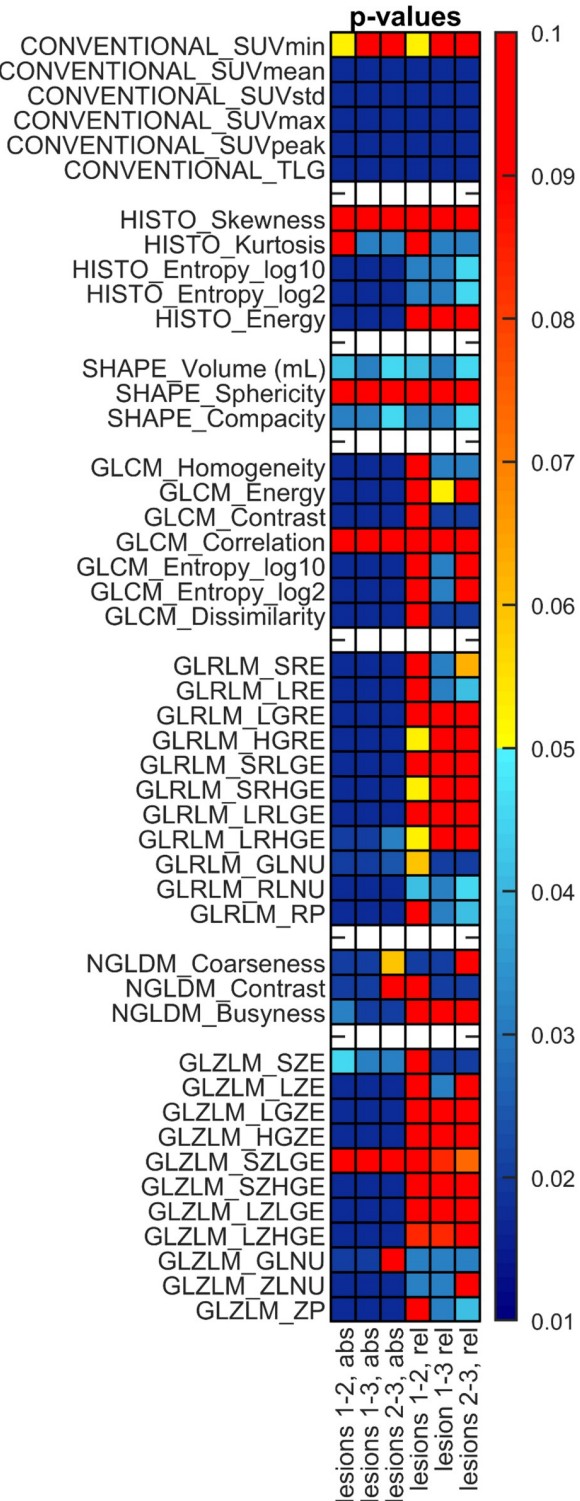

**Fig 8. P-values derived from Wilcoxon signed-rank test.** The colormap contains the *p* values, representing the results of the statistical tests performed between the different lesions, evaluating the data of all eight cameras. Values smaller than p = 5% are considered statistically different.

We found that absolute resampling consistently provided statistically significantly different RIs than relative resampling. Applying the absolute method, most RIs can clearly differentiate between the various lesions. Using relative resampling, however, some GLRLM and GLZLM parameters do not show such a level of distinctiveness, meaning that they cannot be applied in differential diagnostic settings. Regardless of the discretization method employed, no statistically remarkable difference was encountered when comparing the GLCM_Correlation and GLZLM_SZLGE parameters of any lesion.

### Intraclass correlation coefficient

The values of relative and absolute discretization are presented in Fig 9.

We demonstrated that radiomics parameters obtained with absolute discretisation performed far better than the ones acquired using relative discretisation. Based on the results of absolute discretisation, 24 parameters demonstrated good repeatability, out of which 12 had ICC values above 0.9. Although 5 radiomics parameters ($SUV_{mean}$, $SUV_{std}$, $SUV_{max}$, TLG, $SUV_{peak}$) showed good repeatability upon reconstruction with relative discretisation, we experienced poorer overall repeatability with the application of this method. Apart from $SUV_{min}$, the conventional parameters with ICC values above 0.8 were the most repeatable. Lastly, the following higher-order parameters were considered to have good repeatability with absolute resampling as well: GLCM_Homogeneity, GLCM_Energy, GLCM_Contrast, GLCM_Entropy_$log_{10}$, GLCM_Entropy_$log_2$, GLCM_Dissimilarity, GLRLM_SRE, GLRLM_LRE, GLRLM_LGRE, GLRLM_HGRE, GLRLM_SRHGE, GLRLM_LRLGE, GLRLM_RP, GLZLM_HGZE, GLZLM_SZGHE, GLZLM_ZLNU, GLZLM_ZP.

## Discussion

To our knowledge, no previous multi-center studies have applied phantom measurements with such a high reproducibility. Implementing long-lived $^{22}$Na point source, we could guarantee a low change in the radioactivity of the lesions during the simulation. Previous tests have demonstrated that the reproducibility of the PET uptake values using our method (0.68%, 1.57% and 1.52% CV values for $SUV_{mean}$, $SUV_{max}$, and $SUV_{min}$; respectively) exceeds that of the conventionally used NEMA calibration phantoms [31]. It is also worth emphasizing that according to a review paper [26], this method is the only way to simulate arbitrary textural patterns in the PET field. The only limitation was that all activity distributions were created in the air without incorporating the Compton scatter and the attenuation effect.

The current study strengthens the possibility of developing a realistic PET radiomic phantom based on the extension of our previously introduced activity painting method (Fig 1) [31]. Applying the activity painting in a water tank, the greatest challenge was to achieve the lowest possible water waves. For this reason, the movement was programmed so that the speed of the point source between the grid points did not exceed 2 mm/sec. Overall, the waves in the water were minimized by moving the L-shaped rod this way. To ensure easy reproducibility, readily available materials were used for our research.

Retrospective human studies often span a decade from the inception of the study, that results in the shortage of cases utilizing contemporary PET devices and reconstruction methods [41–44]. In contrast, our phantom technique enables measurements to be conducted using the latest imaging technologies. Furthermore, the activity painting technique also aids multi-center studies since, with the proper simulations, potentially different tumor stages and tumor subtypes can be generated. Using the segmented and the simulated lesions of the patients makes texture feature evaluation possible both before and after treatment in a multicentric

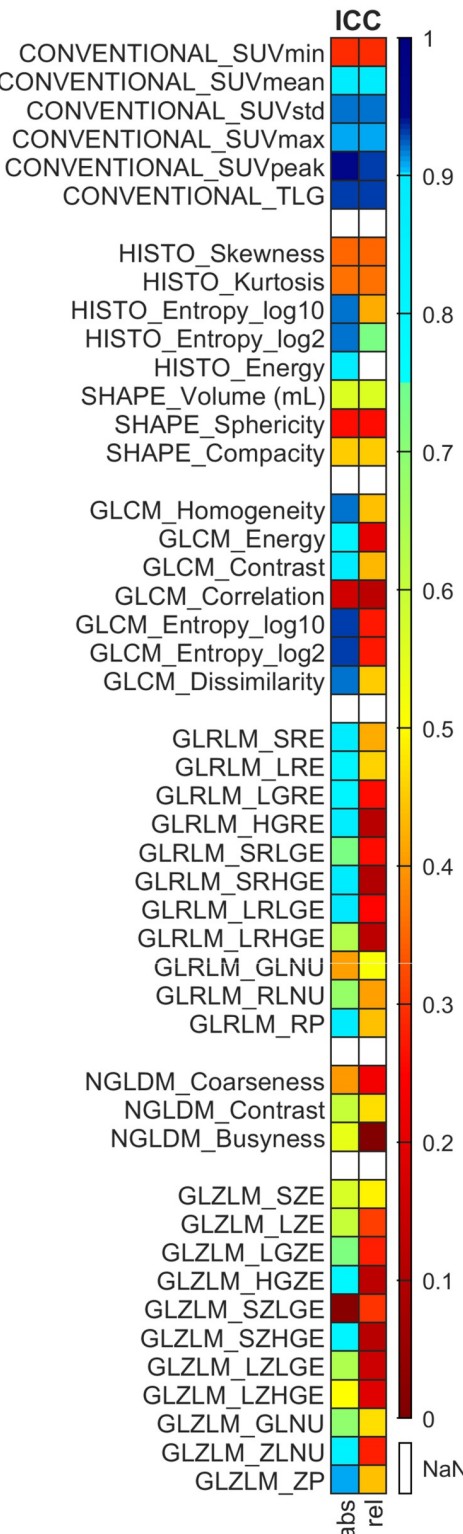

**Fig 9. Intraclass correlation coefficient (ICC) of the calculated radiomics parameters.** Based on the value of the ICC, radiomics indices can be characterized by excellent (ICC>0.9, marked with dark blue), good (0.75<ICC≤0.9, marked with light blue), moderate (0.5<ICC≤0.75, marked with green) and poor (ICC≤0.5, marked with yellow, orange, and red) repeatability. Values marked with white have ICC<0.

environment. Moreover, our realistic lesions could help to address the issue of harmonization,–a task that is often insurmountable with traditional phantoms.

In this study, three lesions were selected from an online database (Fig 2) and simulated on five different PET cameras. Reconstructions were performed using eight distinct settings with different voxel sizes (Table 1).

Upon data assessment, poor reliability and reproducibility were observed for the Volume and TLG, which could be attributed to manual segmentation. Table 1 shows the background activity at the beginning of the simulation, which gradually declined throughout the simulation of all three lesions. By the end of the measurement, the activity concentration of the water decreased to 50% of its initial value. This contrast difference is also manifested in Fig 3, where Lesion 1 displays a clearly detectable rectangular border, Lesion 2 has a less active rectangle around it, while Lesion 3 does not exhibit any visible border at all. During the initial evaluation of the reconstructed images, we endeavored the threshold-based segmentation methods available in LIFEx. However, given the contrast differences between the lesions and the background, we could not identify the same threshold value applicable for all lesions, therefore manual segmentation was applied.

Manual segmentation and discretization were carried out using the LifeX software. LifeX offers multiple segmentation methods; however, these could not be exploited because of the varying backgrounds. After segmentation, the values of 46 radiomic indices were calculated.

The reconstructed images are presented in Fig 3. Comparing these images, it seems evident that although the imitated pattern was the same in case of all acquisitions,–as expected—because of the different characteristics of the applied devices and reconstruction settings, disparities could be detected between the scans [45]. As fewer details were visible using settings with larger pixel sizes, we noted that these tended to blur the image to a greater extent than those with smaller pixel sizes. For example, the images obtained with settings G or H were more detailed compared to those performed utilizing setting B. Out of the selected lesions, the pattern of Lesion 1 was found to be the most compact one. Following its evaluation, no meaningful visual differences were noticed between scanners A, B, C and D; however, using the high-end digital PET system (scanner E), additional details such as a hypometabolic central area or undulated borders could be identified. Taking the above into account, we confirmed that the spatial resolution and the reconstruction voxel size of the PET system greatly influence the pattern of the image.

In the next step, we compared the RI values obtained with imaging setting D with those of all the other investigated settings (Fig 4). Comparing the reconstruction settings or the PET cameras, the slightest differences were observed for the conventional parameters. In case of Lesions 1, 2, and 3, altered RD values were detected for $SUV_{std}$: the RDs of Lesion 3 were smaller than those of the other Lesions. As for the GLCM parameters, we observed that Lesions 1 and 2 typically showed higher RD values in comparison with Lesion 3. A more outstanding dissimilarity could be experienced regarding TLG, presumably due to the variability of the segmented volumes generated by manual segmentation. Some GLRLM-based parameters, including GLRLM_SRE, GLRLM_LRE, and GLRLM_RP, were not sensitive to the different imaging settings. Although, according to previously published data, GLZLM-based parameters were the most reliant on resampling and the different imaging settings [27, 46]. Moreover, we noticed that these higher order parameters resulted in relative differences greater than 25% in almost every case.

Inter-setting COV was used to assess or rank the variability of the RIs between the different imaging settings (Fig 5). Nevertheless, in our case, the CV was determined not on the basis of the conventional and clear test-retest measurement configuration—where exactly the same measurement is repeated several times—but on the basis of the results obtained with the

different imaging settings. Therefore, a small CV value of a given RI could be attributed to two facts: i) first, the RI parameter concerned is robust, i.e. is not sensitive to the measurement procedure, consequently it is a valuable parameter; ii) second, it could also mean that the given RI is not a suitable metric for a PET image, because the RI remains nearly constant even though the texture of the image is changed due to the imaging settings.

In recent works, radiomics features were also ranked by CV analysis, where different PET/CT systems or image processing configurations were the basis of ranking [14, 47]. According to these publications, RIs with smaller CVs were considered to be more reliable or better parameters. Interpreting the CV analysis results of the first order RI parameters, we suppose that a smaller CV projects higher reliability. Prior studies have established that SUV is a highly repeatable parameter, and the within-subject CV is approximately 10% [48]. Rasmussen *et al.* achieved low variability of the different SUV values in a patient study using the Siemens Biograph mCT 64 PET/CT scanner and a Siemens Biograph mMR PET/MR with a 3T magnet (CV5.7% for $SUV_{mean}$, 4.8% for $SUV_{max}$, and 5.7% for $SUV_{peak}$) [49]. Other results on patient and phantom scans demonstrated that the CV of $SUV_{mean}$, $SUL_{peak}$, Volume and TLG parameters were $\leq$5%, and the $SUV_{max}$ was in the range of 5–10%. In our study, the SUV parameters showed larger variability (9.33%, 10.26%, 7.03%, 17.27% and 25.06% CV values for SUVmean, SUVmax, SUVpeak, TLG and Volume; respectively) [50]. The fact that in the previous studies only two different PET cameras were used for the acquisition, while we implemented eight different imaging settings could provide reasonable explanation why higher CV values were detected in the present study in comparison with earlier findings. In addition, $SUV_{peak}$ seemed to be the most robust RI regarding all lesions, imaging, and post-processing methods (as demonstrated in Fig 6). As for the higher order RI parameters, the highest CV values were identified for NGLDM and GLZLM. On the contrary, we observed that GLCM_Entropy, GLRLM_SRE, GLRLM_LRE and GLRLM_RP resulted in low CVs with good overall robustness. Investigating the same parameters, Branchini et. al. also obtained similar results regarding the CV, that was in line with our observation [51].

Our results were in accordance with those of Li *et al.*, who employed the same set of criteria for the CV values as the one used in the present study [46]. The values of some parameters (HISTO_Entropy, HISTO_Energy, GLRLM_SRE, GLRLM_LRE, GLRLM_HGRE, GLRLM_SRHGE, GLRLM_LRLGE, GLZLM_SZE, GLZLM_HGZE, and GLZLM_ZP) showed periodicity by the FBS and FBN discretization methods. Considering that the CV values are highly dependent upon the radiomics parameters, based on our current outcomes, we cannot state that any of the presently applied discretization methods produced better CVs than the other. Although the Image Biomarker Standardization Initiative (IBSI) recommends using relative discretization methods in quantitative imaging systems, recent studies favor the use of FBS method instead of FBN [52–55]. Additionally, in some cases, smaller CVs could be registered for Lesion 3 than for the other two lesions. As for GLCM_Correlation and GLCM_Dissimilarty,–for example—CVs were generally below 10% and at about 20% for Lesion 3 and Lesions 1–2; respectively. Lesion 3 possessed the most complex shape and texture out of the three lesions, therefore, we may assume that those parameters that are sensitive to PET texture are less dependent on the different image settings. This raises the possibility that, for radiomics features, the inter-setting CV number may not be a suitable measure of the reliability of a RI parameter. Furthermore, there is currently no data on the possible RI range for PET lesions of different shapes and textures, though some promising RIs are not considered to be sufficiently sensitive for all ranges (textures).

As shown in Fig 7, spatial resampling may affect the computed RIs as well, aligning with previous results [56, 57]. Considering that spatial resampling is a critical step in the harmonization of measurement settings, evaluating its effects was also our focus of interest.

Accordingly, spatial resampling was performed on the images with the LIFEx software, applying 1 mm, 2.5 mm, 3 mm, and 4 mm voxel sizes. Overall, the CVs of approximately 20% of the investigated RIs were strongly influenced by this step. As expected, the CV of the first order RIs remained unchanged, however, the CV of the imaging settings for the GLCM and NGLDM parameters was substantially affected by spatial resampling. Furthermore, while no CV change was observed for the GLRLM and GLZLM groups, for lesion 3 there were more RIs whose CVs were affected by resampling.

In addition, changing the spatial resampling of the PET data inherently modifies the noise in each voxel. Thus, deviations in RI values related to resampling may be derived not only from changes in pattern structure but also from variations in image noise. Several studies have investigated the relationship between the accuracy of RI and image noise in PET imaging. The repeatability of RIs is highly,–or more precisely monotonically,–dependent on the image count statistics, which can be globally characterized by the total count in a VOI—often referred to as exposure—which is therefore the product of the activity of a given VOI and the acquisition time [27, 29, 58–60]. Literature data also report on the availability of such optimal exposure value range where the RI values are exposure invariant (no change in the RI value within that range), regardless of the image reconstruction algorithm and the selected voxel size [58].

In technical terms, the injected activity and scan length must be chosen in a way that the RIs are robust to image noise, meaning that it might result in only minimal variability and bias. This way, textural features become comparable across nuclear medical sites using different PET/CT systems, which could lay the basis for the usage of texture metrics in multi-center studies [58, 60]. Somasundaram *et al.* further demonstrated that radiomics features can be corrected for noise-induced bias using the local STD or CV of the voxel values in the VOI of the lesion. The basis of the correction is the preliminary measurement of the noise-based distortion of the RI and the application of a simpler analytical model [59].

An additional noise-based image harmonization technique based on equalizing the contrast-to-noise ratio of images obtained from different imaging and reconstruction protocols was also proposed [27]. Another important fact is that the reproducibility and the reliability of the RIs can significantly vary between homogeneous objects of different sizes [61], as well as between heterogenous and homogeneous VOIs [29]. This is of crucial importance as most studies investigating the relationship between radiometric indices and noise used homogeneous phantoms, while with our newly established activity painting method arbitrary heterogeneous shapes can also be simulated.

Wilcoxon signed-rank tests were performed to analyze all RI values of the different lesions (Fig 8). Corresponding to the results, absolute resampling provided statistically significantly different RIs than relative resampling. Using absolute resampling, most of the RIs were capable to clearly distinguish between the various lesions thus, they ensured definitive lesion identification. Nevertheless, with the application of relative resampling, several GLRLM and GLZLM parameters did not show such level of distinctiveness, indicating that they cannot be safely used for differential diagnostic purposes.

Regarding the ICC, the radiomics parameters obtained with absolute discretization had much higher values than those acquired using relative discretization (Fig 9). As for absolute discretization, 33 parameters showed improved repeatability, out of which 21 presented ICC values greater than 0.9, indicating excellent repeatability. In contrast, lesions reconstructed with relative discretization were assumed to have worse repeatability in general. Evaluating the influence of different discretization methods on the ICC, Branchini *et al.* also published that absolute discretization methods yielded notably elevated ICC values, and this corresponded to our observation [62]. Although, in their study, a total of 55 higher order parameters were included, and among those 45 had excellent repeatability with absolute discretization. While

we used eight different imaging settings, Branchini and colleagues employed only one PET device, and this methodological difference could underpin the experience of better repeatability in their study.

The conventional parameters with ICC above 0.8 appeared to be the most repeatable. As for higher order parameters worse repeatability was observed; however, the ICC values of the radiomic parameters were influenced by the contouring variability derived from manual segmentation [63]. According to our results, it could be highlighted that absolute discretization generated greater differences, hence harmonization may be a more critical issue in that case.

Generally, the visual differences between the images of the lesions (resulting from different imaging settings) are expected to be reflected at the level of textural parameters. Therefore, the fact that the values of a given higher order RI parameter are different (i.e. CVs are greater) for the different imaging settings (and correspondingly different image content) does not necessarily mean that the RI parameter is inappropriate for texture characterization. Indeed, parameters (e.g. SHAPE_Sphericity, GLRLM_SRE) with values without any significant differences between the images (Fig 4) could be considered unsatisfactory.

Another aspect is that cohorts in radiomics analyses are becoming increasingly multicentric, and a vast array of embedded statistical and artificial intelligence methods are available for the selection of appropriate features. This situation raises the question of whether experiments with phantoms bear added value. Based on the above detailed results, we firmly believe that phantom studies are worthwhile to execute, as such measurements of realistic textures can be used to preselect RI parameters with sufficient reliability and texture measurement ability (sensitivity). In addition, knowledge gaps remain to be uncovered regarding the extent to which the RI parameters of lesions with different shapes and textures differ or how this change if the morphology of the given lesions changes, for example, due to disease progression [64]. These fundamental questions can only be fully elucidated by the simulation of textures with exact details followed by PET measurements and analyses. According to the current observations, this may be achieved with our newly established activity painting method.

Even though our measurements enable the proposal of a highly reproducible heterogeneous phantom, some limitations of the present study must be addressed. First, since only one scan was run for all lesions, the effects of repositioning were not tested. Second, our phantom measurement setup does not consider the differences between the image acquisition protocols (including the injected activity and time per bed position) routinely used in standard human diagnostics. Furthermore, given that the current activity painting system can only interpret a 5cm×5cm×5cm matrix, the size of the lesion to be simulated had to be adjusted to this predefined volume. Finally, the acquisition time (20–30 minutes) and the preparation duration resulted in an approximately 3-hour-long measurement time.

## Conclusion

Comparing five PET cameras on the basis of three simulated tumor patterns, we managed to confirm that absolute discretization is more suitable for diagnostic purposes than the relative method, and that spatial resampling seems to be crucial for higher order features. Further, conventional SUV parameters proved to be more reliable, while size-zone and run-length-based parameters demonstrated lower reliability. It was also found that inter-setting CV was not an appropriate metric for analyzing the reliability and the robustness of the RI parameters. Overall, our novel activity painting phantom technique ensures the performance of multi-center studies by analyzing the effect of imaging protocols on radiomics features. Furthermore, it not only facilitates pre,–and post-treatment lesion evaluation without the necessity of actual

patient involvement, but also allows for the use of older and state-of-the-art PET imaging technologies and thus the assessment of image harmonization.

## Supporting information

**S1 Table. List of radiomics indices.** The parameter group, the abbreviation and the full name of the corresponding textural indices are also ennumerated in the table.
(XLSX)

## Acknowledgments

The authors would like to express special thanks to Jozsef Molnar (Institute for Nuclear Research, Debrecen) for the thoughtful conversations and the helpful support during the period of this work.

## Author Contributions

**Conceptualization:** Piroska Kallos-Balogh, Attila Forgacs, Laszlo Balkay.

**Data curation:** Piroska Kallos-Balogh, Norman Felix Vas.

**Formal analysis:** Lilla Szatmáriné Egeresi.

**Investigation:** Piroska Kallos-Balogh, Norman Felix Vas, Zoltan Toth, Szabolcs Szakall, Peter Szabo, Ildiko Garai, Attila Forgacs, Lilla Szatmáriné Egeresi.

**Software:** Piroska Kallos-Balogh, Norman Felix Vas.

**Supervision:** Dahlbom Magnus, Laszlo Balkay.

**Validation:** Piroska Kallos-Balogh.

**Visualization:** Piroska Kallos-Balogh, Norman Felix Vas.

**Writing – original draft:** Piroska Kallos-Balogh, Zoltan Toth, Szabolcs Szakall, Peter Szabo, Ildiko Garai, Zita Kepes, Attila Forgacs, Laszlo Balkay.

**Writing – review & editing:** Piroska Kallos-Balogh, Zita Kepes, Lilla Szatmáriné Egeresi, Dahlbom Magnus, Laszlo Balkay.

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
