## [Decision Letter · Decision Letter 0]

15 May 2024

PONE-D-24-08717Multicentric study on the reproducibility and robustness of PET-based radiomics features with a realistic activity painting phantomPLOS ONE

Dear Dr. Kallos-Balogh,

Thank you for submitting your manuscript to PLOS ONE. After careful consideration, we feel that it has merit but does not fully meet PLOS ONE’s publication criteria as it currently stands. Therefore, we invite you to submit a revised version of the manuscript that addresses the points raised during the review process.

We look forward to receiving your revised manuscript.

Kind regards,

Fei Yang

Academic Editor

PLOS ONE

Journal Requirements:

4. Please upload a copy of Supporting Information Figure/Table/etc. "supplementary table" which you refer to in your text on page 10.

**Additional Editor Comments:**

This study investigated the reproducibility and robustness of PET-based radiomics parameters using a realistic activity painting phantom. While the study holds potential significance, as pointed out by the reviewers, the methodology used and the presentation of key findings exhibit several deficiencies and warrant further work. Furthermore, a number of studies have explored the use of digital phantoms to assess the validity of PET radiomics and to evaluate the stochastic effects resulting from different acquisition and processing parameters on PET radiomics. The authors are encouraged to review this body of work and to contextualize their findings within the existing research landscape on this topic. In addition, the quantity of simulated lesions is rather restricted, potentially resulting in low study power. The authors are advised to augment the number of lesions, as doing so would not only improve the study's statistical power but also allow for a more precise representation of heterogeneous and highly variable clinical tumors.

Reviewers' comments:

Reviewer's Responses to Questions

**Comments to the Author**

1. Is the manuscript technically sound, and do the data support the conclusions?

Reviewer #1: Yes

Reviewer #2: Partly

2. Has the statistical analysis been performed appropriately and rigorously? 

Reviewer #1: Yes

Reviewer #2: No

3. Have the authors made all data underlying the findings in their manuscript fully available?

Reviewer #1: Yes

Reviewer #2: No

4. Is the manuscript presented in an intelligible fashion and written in standard English?

Reviewer #1: Yes

Reviewer #2: No

5. Review Comments to the Author

Reviewer #1: The manuscript aims to investigate an "activity painting" tool for PET image simulation simulate arbitrary lesions in a radioactive background to perform relevant multicenter radiomic analysis. The paper is of interest to readers, and it is presented in a precise and concise fashion. I only suggest to improve the quality of the lesions images, as it is currently very poor.

Reviewer #2: Dear authors,

The topic of the manuscript is of utmost importance; however, there exist clear technical issues and the manuscript should be carefully reviewed.

Please see the attached pdf with further comments that might help improve the overal quality of the manuscript, taking care of specific points that are not 100% clear, as well as some statistical concerns.

Additionally, even if the overall English is good, the flow of the manuscript is sometimes erratic and lacks connection between paragraphs. I suggest the authors to review the text to carefully report their findings.

Again, please see the attached document for further comments.

Thank you.

6. PLOS authors have the option to publish the peer review history of their article (what does this mean?). If published, this will include your full peer review and any attached files.

Reviewer #1: No

Reviewer #2: No

---

## [Author Response · Author response to Decision Letter 0]

13 Jul 2024

Dear Editor and Reviewers,

We would like to express our sincere gratitude for your thoughtful and constructive feedback on our manuscript titled " Multicentric study on the reproducibility and robustness of PET-based radiomics features with a realistic activity painting phantom". We appreciate the time and effort and your attention to detail invested in reviewing our work, and we are grateful for the opportunity to improve our manuscript based on your insightful comments.

We have carefully considered each of your suggestions and have made the following responses and revisions to address your concerns:

Reviewer 1: 

The manuscript aims to investigate an "activity painting" tool for PET image simulation simulate arbitrary lesions in a radioactive background to perform relevant multicenter radiomic analysis. The paper is of interest to readers, and it is presented in a precise and concise fashion. I only suggest to improve the quality of the lesions images, as it is currently very poor.

Thank you for your positive feedback on our manuscript. We have taken it into consideration and have made significant improvements to the images of the lesions, particularly in Figure 2 and Figure 3. We hope these changes meet your expectations and enhance the clarity and quality of the visual representations in our paper.

Reviewer 2:

The authors of this manuscript focus on investigating a multicentric study on the reproducibility and robustness of PET-based radiomics features with a realistic activity painting phantom. As they state, there exist certain variability on the obtention of PET radiomic features, as well as in the associated radiomic pipelines. Working towards harmonization of PET images and homogenization of the pipelines is desirable, but there still exist different factors hindering the establishment of reproducible and robust PET radiomic pipelines. Their methods and findings underscore the need for such standardization in PET radiomics. 

The topic of the manuscript is of interest in the community; however, the manuscript frequently lacks clarity and is difficult to follow. Considering a few points would improve the overall quality of the manuscript and improve its content: 

1. The ideas presented in the introduction seem a bit isolated and, sometimes, it is difficult to read and follow, with the connection between paragraphs is a bit weak; I would suggest the authors to revisit the introduction, extend it, and improve its overall flow.

We have carefully revised the introduction to address your concerns. Specifically, we have reorganized the content to ensure a more logical progression of ideas, enhancing the coherence and flow between paragraphs. The transitions between the paragraphs have been improved to create a smoother narrative and facilitate better understanding for the reader. 

2. The authors should be more careful when reporting data. For example, in table 1, there are values reported with different resolutions (i.e., volume, or time inaccuracy). Please check for consistency.

The text has been thoroughly checked for consistency and accuracy. We have paid special attention to the units of time and volume and consistently given them in second and Liter (L or mL) wherever they occur. 

3. Figure 2 should describe the actual orthogonal orientations of the images.

To address your comment, we updated the figure, which includes the orthogonal views, clearly depicting the actual orientations of the images. Additionally, we have ensured that the images are presented without smoothing to make them easier to recognize and interpret. 

4. The authors write several times about the measurements being carried out with the application of the PET scanners. This is a bit weird to say, as, even if PET counts are measured, images are acquired, or better said, data is acquired and images reconstructed, using the PET scanners.

Thank you for pointing out the ambiguity of this expression. We have revised the relevant sentences to reflect the process more accurately. Specifically, we have changed the wording to indicate that PET scanners collect data and then reconstruct the images. 

5. In table 2, the authors should better describe the contents of the different cells. Not everyone reading the paper should be able to understand the reconstruction settings, regarding algorithms (OSEM, TT3D, etc.) and their parameters (i.e., iterations, subsets, etc.). Table caption should, at least, include the definition of the acronyms and the format of number of iterations and number of subsets.

Thank you for your recommendation on Table 2. We have made the following revisions:

• Added definitions: The definitions of the following abbreviations have been included in the table caption: 2D OSEM, FWHM, TT3D, VUE Point HD, 3D RAMLA, QClear, SharpIR, VPFXS.

• Clarified Reconstruction Settings: We have improved the descriptions of the contents within the cells to provide a clearer understanding of the reconstruction algorithms and their parameters. This includes specifying the format for the number of iterations and number of subsets, as well.

We believe the revised table caption now provides comprehensive definitions of the acronyms, facilitating better comprehension of the reconstruction settings.

6. What is a “standard” reconstruction here?

The term "standard" was indeed incorrect as it was only the name of the reconstruction protocol. We have revised it, replacing the "standard" word with the correct reconstruction settings.

7. Codes C, D, E and F are reported with no filtering. In which scenarios is data not filtered and smoothed? Most clinical data are smoothed in some way, so I am wondering how close to the reality those post-processing steps are.

As you suggested, we have thoroughly reviewed Table 2 and incorporated the previously missing information. The table now accurately presents the details regarding scenarios where data is not filtered or smoothed. Specifically, we have clarified that the image reconstruction processes of Mediso AnyScan PET/CT (C) and Philips Gemini TF 64 (F) systems did not employ post-filtering. 

8. It is unclear how the authors calculated the radiomic features. In-house? Any package?

The calculation process is described on page 10 of the revised manuscript. We used the freely available LIFEx v4.34 software to ensure standardized and reproducible extraction of radiomic features.

9. The authors compute first order statistics, neglecting the description of histogram and intensity radiomic features such as skewness, kurtosis, energy, etc. which are commonly used in radiomic approaches, but then are included in figure 4. They should better describe how they actually compute the radiomic features and which ones they are including in the study.

On page 11 we now enumerate all first order parameters, including histogram and intensity-based features. Also, these parameters are mentioned several times throughout the results and discussion section, and the specific features you highlighted are often separately emphasized. We are aware, that the number of histogram features included is limited, but these are the ones supported by the LIFEx v4.34 software.

10. Apart from those and high order features, the authors may want to discuss the use of transformations and radiomic features derived from those transformations.

We appreciate your comment. However, it was out of our scope to investigate the relationship between image transformations (such as wavelet tr. or Gabor filters) and the derived features. Our study intends to provide a new phantom application for the radiomic community that ensures the generation of reproducible textures for analysis of radiomic features and their reliability. We believe that based on this method, the effect of the image transformations will be able to investigate in future studies.

11. Also, the nature of the phantom and the whole work neglect the inclusion of the shape features, which are agnostic and more robust than the rest of features… But again, shape features are included in figure 4 (despite being just a few). This should be better described, noted and discussed by the authors

We have revised the manuscript to address your concerns. On page 11, we now enumerate all first-order and shape parameters. Throughout the results and discussion sections, these parameters are mentioned several times. We are aware, that the number of shape features included is limited, but these are the ones supported by the LIFEx v4.34 software.

12. In the statistical analysis, the authors chose the D imaging setting arbitrarily; however, I am not sure if that was a good decision, as that option uses the “standard” unknown method and has no post-processing, which implied it might be more contaminated by noise.

We agree with Your comment. There was an error in Table 2 previously, which we have now corrected. After this revision, we can confirm that this setting also included a filter. At the time of the study, we chose an average camera to serve as the standard, but this choice was arbitrary, not of great significance.

13. Figure 3 lesions do not seem to match lesions from figure 2. Please explain and include the lesions from figure 2 as a main row at the top of figure 3. Also, include a colorbar showing the actual values of the PET uptake (are those just MBq or SUV?).

We have modified Figure 3 to include the lesions from Figure 2 as the last row. Additionally, we have added a SUV colorbar to indicate the actual values of PET uptake. 

Regarding the observed discrepancies, it is important to note that the simulated lesions will not appear exactly the same compared to the original lesions, due to the varying imaging properties of different cameras. 

However, despite these variations, Figure 3 allows us to clearly identify which lesions originate from the same initial lesions, and they match the original lesions to a certain extent. We believe this level of matching demonstrates the consistency of our simulation technique while accounting for the inherent differences in imaging properties.

14. Figure 4 does not only show relative, but also absolute, differences in features. This seems to be inconsistent with the text. Also, why there is no 0 value? If two methods are equivalent, that value should be considered… Additionally, relative values could be negative. These results should be revisited and better reported/discussed.

Considering your comment, we have revised the colorbar of Figure 4 to appropriately reflect the differences and ensure it now includes the 0 value. No two methods are equivalent, so the values will never be 0. The D method was excluded to avoid redundancy, because comparing method D with itself will of course, result in zero values. The changes are now discussed in the figure's description to provide clarity. Although relative differences can be negative, as described in the Materials and Methods section, the relative difference calculation in our statistical analysis uses absolute values. 

15. Figure 6 shows the same trend and potential error, as all abs and rel values are the same! Are these values correct or is the error always biased on the same direction?

We can confirm that the results presented in Figure 6 are correct. Histogram-based parameters are not sensitive to the different discretization settings, which is why the absolute and relative values appear the same.

16. How was the Wilcoxon signed-rank test run? Did the authors consider the number of tests and correct for multiple comparisons? How? Given the great number of tests, there could be some statistical results that are just derived from random results.

Given the number of tests conducted, we appreciate your detailed concern about the necessary correction of p-values for multiple comparisons. Indeed, we did not perform this type of correction; thus, we included this calculation in the revised work. The Benjamini-Hochberg false discovery rate (FDR) method was used to correct the P-values calculated using the Wilcoxon signed-rank test. The FDR-based control is less stringent than the Bonferroni method, however it greatly improves the power of statistical inference. The updated Figure 8 shows slightly, but not substantially, different p-values from the unadjusted p-values, so we did not need to change the main inferences on the hypothesis tests in the discussion part. 

17. Conclusions should include a better summary of the contribution to increase the overall soundness.

We have revised the conclusions section to provide a clearer and more comprehensive summary of our contributions. The updated conclusions now highlight the key findings of our study, their implications for the field, and how they advance current knowledge. We believe this enhanced summary increases the overall soundness and impact of our manuscript.

Additional Editor Comments:

This study investigated the reproducibility and robustness of PET-based radiomics parameters using a realistic activity painting phantom. While the study holds potential significance, as pointed out by the reviewers, the methodology used and the presentation of key findings exhibit several deficiencies and warrant further work. Furthermore, a number of studies have explored the use of digital phantoms to assess the validity of PET radiomics and to evaluate the stochastic effects resulting from different acquisition and processing parameters on PET radiomics. The authors are encouraged to review this body of work and to contextualize their findings within the existing research landscape on this topic. In addition, the quantity of simulated lesions is rather restricted, potentially resulting in low study power. The authors are advised to augment the number of lesions, as doing so would not only improve the study's statistical power but also allow for a more precise representation of heterogeneous and highly variable clinical tumors.

Thank you for your detailed and constructive comments. We have reviewed the literature on digital phantoms and PET radiomics, which investigated the stochastic effects on RI from different PET parameters. In the revised discussion section, we have included seven additional relevant works that explore the role of noise on radiomics analysis.

We acknowledge your concern about our study's limited number of simulated lesions. However, the main goal of our work was to demonstrate the feasibility of applying a novel phantom technique to simulate arbitrary heterogeneous patterns in background activity. While we understand that increasing the number of simulated lesions would enhance the statistical power of the study, our focus in this initial investigation was to establish the feasibility of the technique. We plan to explore this aspect further in future studies and appreciate your understanding of the practical constraints involved in this preliminary work.

---

## [Decision Letter · Decision Letter 1]

14 Aug 2024

Multicentric study on the reproducibility and robustness of PET-based radiomics features with a realistic activity painting phantom

PONE-D-24-08717R1

Dear Dr. Kallos-Balogh,

We’re pleased to inform you that your manuscript has been judged scientifically suitable for publication and will be formally accepted for publication once it meets all outstanding technical requirements.

Kind regards,

Fei Yang

Academic Editor

PLOS ONE

Additional Editor Comments (optional):

Reviewers' comments:

Reviewer's Responses to Questions

**Comments to the Author**

1. If the authors have adequately addressed your comments raised in a previous round of review and you feel that this manuscript is now acceptable for publication, you may indicate that here to bypass the “Comments to the Author” section, enter your conflict of interest statement in the “Confidential to Editor” section, and submit your "Accept" recommendation.

Reviewer #1: All comments have been addressed

2. Is the manuscript technically sound, and do the data support the conclusions?

Reviewer #1: Yes

3. Has the statistical analysis been performed appropriately and rigorously? 

Reviewer #1: Yes

4. Have the authors made all data underlying the findings in their manuscript fully available?

Reviewer #1: Yes

5. Is the manuscript presented in an intelligible fashion and written in standard English?

Reviewer #1: Yes

6. Review Comments to the Author

Reviewer #1: After a careful and detailed improvement following the reviewers' comments, the paper can now be considered for publication in PLOS ONE.

7. PLOS authors have the option to publish the peer review history of their article (what does this mean?). If published, this will include your full peer review and any attached files.

Reviewer #1: No

---

## [Editor Report · Acceptance letter]

19 Aug 2024

PONE-D-24-08717R1 

PLOS ONE

Dear Dr. Kallos-Balogh, 

I'm pleased to inform you that your manuscript has been deemed suitable for publication in PLOS ONE. Congratulations! Your manuscript is now being handed over to our production team.

Kind regards, 

on behalf of

Dr. Fei Yang 

Academic Editor

PLOS ONE